# Estimation of Cotton Nutrient Uptake Based on the QUEFTS Model in Xinjiang

Halihashi Yibati [1], Yan Zhang [1,*], Qingjun Li [1], Xingpeng Xu [2] and Ping He [2]

[1] Institute of Soil, Fertilizer and Agricultural Water Conservation, Xinjiang Academy of Agricultural Sciences (XAAS), Urumqi 830091, China; harlhax10@163.com (H.Y.); gyqc@163.com (Q.L.)
[2] Institute of Agricultural Resources and Regional Planning, Chinese Academy of Agricultural Sciences (CAAS), Beijing 100081, China; xuxinpeng@caas.cn (X.X.); phe@caas.cn (P.H.)
* Correspondence: yanzhangxj@163.com

**Abstract:** The Xinjiang cotton-producing region is the main cotton-growing region in China; however, the combination of excessive application but low utilization of fertilizers has restricted its agricultural development. Estimating the balanced nutrient requirements for cotton is essential to increase its yield and nutrient use efficiency. The results from 372 field experiments performed between 1996 and 2019 were used to build a QUEFTS (quantitative evaluation of the fertility of tropical soils) model to investigate relationships between cotton seed yield and nutrient accumulation, with the data from 2017 to 2020 used to validate the model. The results demonstrated that the QUEFTS model predicted a linear relationship between target yield and nutrient uptake until the target yield reached about 60–70% of the potential yield. To produce 1000 kg of seed cotton, 28.3 kg N, 6.1 kg P, and 29.6 kg K were required for cotton, and the corresponding ratio of N, P, and K was 4.64:1:4.85. The corresponding internal efficiencies (IEs) for N, P, and K were 35.4, 163.2, and 33.7 kg/kg, respectively. Field validation indicated that the QUEFTS model could be used to estimate nutrient uptake at a targeted yield and contribute to developing a fertilizer recommendation strategy for Xinjiang cotton production.

**Keywords:** cotton; seed yield; QUEFTS model; nutrient uptake; internal efficiency (IE)

## 1. Introduction

Globally, cotton (*Gossypium hirsutum* L.) is the most important fiber crop. It significantly improves the economies of many countries through the provision of fiber, oil, and several other products [1]. China is the top cotton producer in the world [2], with the Xinjiang cotton region being the largest cotton-growing region in terms of yield, total production, and planting area [3].

The Xinjiang cotton region accounts for 54.0% of the entire cotton planting acreage and more than 67.3% of the total national output [4]. A balanced fertilization program that takes into account organic and inorganic nutrients should be one of the most important factors in cotton production for high yield and quality [5,6]. To pursue higher yield and greater economic benefit, Xinjiang farmers often input excessive nitrogen (N) and phosphate (P) fertilizers while ignoring potassium (K) fertilizer, resulting in the disproportion of NPK. This is associated with negative impacts on yields and the environment, including fertilizer residual in the soil, water pollution, and health risks [7–9]. Therefore, a science-based fertilizer recommendation method must be established to maximize the cotton yield and reduce environmental risk.

Estimating the nutrient requirements of cotton is essential for rational fertilization [10], but fertilization based on limited experimental data has been unable to meet the growing requirements because there was high diversity in soil fertility, climate condition, nutrient supply, and cotton varieties [11]. Several methods of fertilizer recommendation, such as the use of leaf chlorophyll meters [12,13], the technique of petiole N analysis [14], the

diagnosis and recommendation integrated system (DRIS) [15], and soil analysis and yield target [5], have all been applied to cotton nutrient management. These methods, while effective for fertilizer recommendation, generally only involve a single nutrient, and the interactions among N, P, and K are not considered. As soil and plant chemical analyses are both time-consuming and expensive, regional-scale fertilization has generally proceeded without them.

Recently, the quantitative evaluation of the fertility of tropical soils (QUEFTS) model has been successfully applied to various crops, providing recommendations for fertilizer application for a number of countries and covering a variety of soil types and agricultural environments. The QUEFTS model was modified by Smaling and Janssen (1993) [16] to estimate the nutrient requirements for a specific yield target and to take into account the climate-adjusted, season-specific yield potential [17]. The QUEFTS model uses a large quantity of data and takes into full account the interactions between N, P, and K, thus avoiding the problems related to using a small amount of data for guiding fertilization [18,19]. In the model, two linear borderlines are utilized to define the range between maximum nutrient accumulation and maximum nutrient dilution in the plant. These envelopes are arithmetically integrated with a linear–parabolic–plateau curve to predict the relationship between crop yield and plant nutrient uptake in the above-ground dry matter [18] and to estimate nutrient management. Therefore, it provides a practical tool for the application of site-specific nutrient management and fertilizer recommendations for major crops [20,21]. The QUEFTS approach has been employed to estimate crop nutrient requirements in studies on wheat [21–24], maize [25–31], rice [32,33], soybean [34,35], sweet potato [36], tea [37], and radish [38], but not yet for cotton.

We conducted field experiments with different climates, soil types, and varieties of cotton plants from 1996 to 2019 to construct the model, and the data from 2017 to 2020 were used to validate the QUEFTS model. The objectives of the present study were to: (1) develop functions that describe the relationships between the cotton seed yields and the above-ground nutrient uptake; (2) estimate the balanced requirements of N, P, and K for cotton using the QUEFTS model; and (3) evaluate the N, P, and K uptake of cotton simulated by the QUEFTS model through field experiments.

## 2. Materials and Methods

### 2.1. Data Source

The database used in this study included 372 field experiments conducted by the China Soil Testing Program, International Plant Nutrition Institute (IPNI) China Program, published or unpublished datasets by our group, and related published field experiments in the peer-reviewed journal of China Knowledge Resource from 1996 to 2019 (Table 1).

**Table 1.** Climate and soil characteristics of cotton experimental sites in Xinjiang, China.

| Regions | District | Climate Type | PH | Organic Matter (%) | Alkali-Hydrolyzable N (mg/kg) | Olsen-P (mg/kg) | NH4OAc-K (mg/kg) | Case (n) |
|---|---|---|---|---|---|---|---|---|
| Southern Xinjiang | Akesu | extreme arid region | 7.8~8.9 | 1.2~23.1 | 20.8~177.9 | 0.7~63.1 | 66.0~349.0 | 134 |
| | Bazhou | | 7.5~8.6 | 0.5~18 | 6.0~87.0 | 2.5~43.0 | 56.0~297.0 | 47 |
| | Kashi | | 7.8~8.2 | 8.6~18.2 | 35.0~113.0 | 1.8~29.7 | 83.0~226.0 | 17 |
| Northern Xinjiang | Hami | arid region | 7.9~8.6 | 9.7~30.9 | 32.0~150.0 | 7.0~34.0 | 51.0~260.0 | 17 |
| | Bozhou | | 7.7~8.7 | 6.0~42.7 | 30.3~212.8 | 3.0~55.1 | 32.0~653.0 | 44 |
| | Changji | | 7.3~8.3 | 5.0~67.2 | 17.5~278 | 3.2~34.2 | 8.1~432.0 | 75 |
| | Tacheng | | 7.7~8.4 | 1.0~30.5 | 8.2~31.0 | 8.2~31.0 | 110.0~260.0 | 38 |

The experimental sites contained variable nutrient management practices commonly used by Xinjiang farmers, establishing a wide range of nutrient dilution and accumulation situations, including farmers' practices, optimal nutrient practices, different fertilizer rates

treatments, and nutrient omission treatments covering 18 cotton-growing counties and 61 cotton varieties in the Xinjiang region of China (Figure 1).

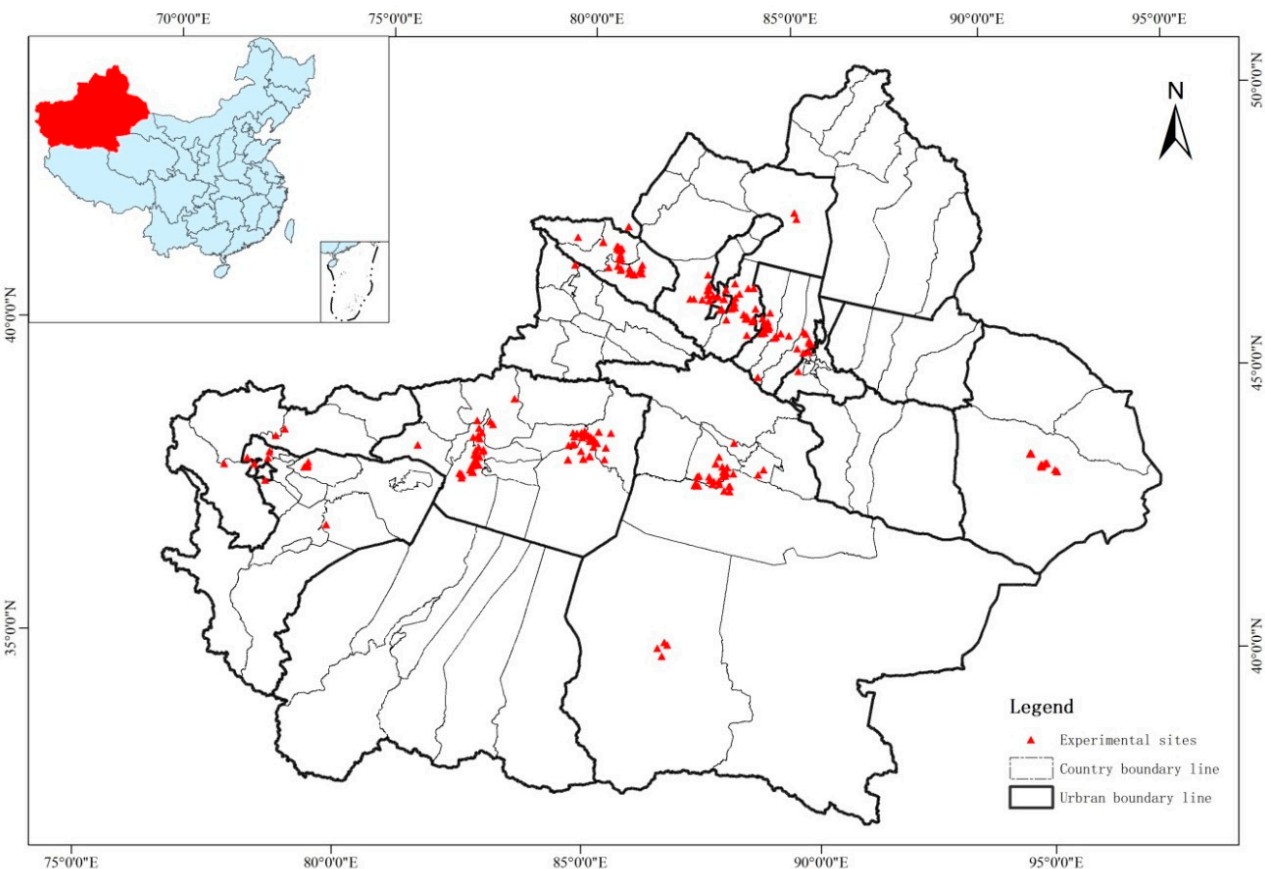

**Figure 1.** Distribution of experimental sites for cotton in different production regions in Xinjiang, China.

After the data collection and arrangement, we analyzed the cotton seed yield, straw yield, harvest index (HI, kg cotton seed dry matter per kg above-ground dry matter), nitrogen, phosphorus, and potassium uptake by the seed cotton and straw, internal efficiency (IE, cotton seed yield per unit of nutrient uptake in the above-ground parts), and reciprocal internal efficiency (RIE, kg nutrient uptake per 1000 kg cotton seed yield) in Xinjiang.

*2.2. Model Development*

The QUEFTS model was originally proposed by Janssen et al. (1990) [19] to evaluate the fertility levels of native and tropical soil. Further improvements were made by Smaling and Janssen (1993) [16] to estimate the nutrient requirements for crop yield. The model takes into account the relationship between N, P, and K rather than the demand for individual nutrient elements alone. The maximum accumulation (a) and maximum dilution (d), which defined the relationships between yield and nutrient uptake, were calculated by excluding the upper and lower 2.5th, 5.0th, and 7.5th percentile of all measured IE. These *a* and *d* values were then used as core parameters to estimate the nutrient requirements of cotton through the application of the QUEFTS model.

The steps of the QUEFTS model have been described previously [18,24,25]. Additionally, the nutrient uptake under different potential and target yields was estimated using the solver model in Microsoft Office Excel. The key steps were, briefly, as follows: (a) selecting appropriate data that meet the boundary conditions of the model; (b) defining the two border lines for N, P, and K corresponding to the maximum and minimum nutrients accumulation in plants; (c) simulation of optimal N, P and K absorption curves at different

potential or target yields; and (d) field validation of the applicability of the QUEFTS model for simulating the cotton nutrient uptake requirements in Xinjiang.

### 2.3. Field Validation

The field experiments were conducted at 14 sites across Xinjiang, including Changji, Akesu, Kashi, Tacheng, and Bazhou regions, from 2017 to 2020, to validate the QUEFTS model. Cotton was planted following the cultivation mode of plastic film mulching, short rod dense planting. Local popular varieties of cotton were planted in each field and were cultivated with optimized management practices. Cotton seeds were generally sown in April and harvested in late September or early October. The plot sizes of each experiment were 43.2 m$^2$ and distributed in a randomized block design. Each plot was mulched with four sheets of plastic film, and two drip irrigation lines were installed under one of the sheets. Each film covered six rows of cotton plants, and the row spacing configuration was 10–66–10–66–10 cm. The cotton plants were sown at 10 cm intervals within each row.

The fertilizer recommendations were provided by the Nutrient Expert System (NE). The NE is a nutrient decision support system developed by the International Plant Nutrition Institute based on an improved SSNM (site-specific nutrient management) nutrient management and QUEFTS models for determining the optimal nutrient requirements of the cotton [39,40]. In the Nutrient Expert, the fertilizer recommendation for N was determined by the yield response (yield gaps between the plot that received ample N, P, and K and omission plot from which one of the nutrients was omitted) and agronomic efficiency of N fertilizer application. The P and K rates were determined by yield response combined with the RIE and the nutrient balance was required to sustain the soil fertility simulated by the QUEFTS model. The treatments included a balanced NE (fertilizer application based on Nutrient Expert) and NE (10% OF) (10% replacement of chemical N fertilizer with organic N fertilizer). The rates of fertilizer application ranged from 216–260 kg N/ha, 102–159 kg P$_2$O$_5$/ha, and 80–134 kg K$_2$O/ha, respectively. Urea (46% N) was used as N fertilizer via drip irrigation in five applications (10:20:30:20:20) over the course of the cotton growing season. P fertilizer was applied 70% Superphosphate (46% P$_2$O$_5$) before sowing and 30% Monoammonium phosphate (12% N, 60% P$_2$O$_5$) during the ray period. K fertilizer potassium sulfate (50% K$_2$O) was applied 50% at the ray stage and 50% at the flowering stage.

At harvest, uniform and robustly growing areas of 9.6 m$^2$ were randomly selected from three fields for each treatment. The number of mature cotton bolls per plant in each treatment area was recorded, from which the number of bolls per plant and the number of plants per unit area were calculated. The lower, middle, and upper 30, 50, and 20 opening bolls were collected in each plot, and the average boll weight and lint percentage were measured to calculate the cotton yield. The above-ground parts of cotton plants were separated from the stem base and underground parts, and the surface dust was removed. The plants were separated according to different organs of straw (stem, leaf, and cotton shell), cotton fiber, and cotton seed and then placed in an oven at 105 °C for 1 h. The samples were then dried to constant weight at 75 °C and then weighed after cooling. The individual plant parts were digested with H$_2$SO$_4$-H$_2$O$_2$, and the concentrations of N, P, and K were determined using the Nissl colorimetry, standard vanadomolybdate, and flame spectrophotometers method, respectively [41]. The accumulation of N, P, and K in the total plant were calculated by multiplying the dry matter weight and nutrient concentration.

### 2.4. Statistical Analysis

The value of the RMSE and n-RMSE were used to evaluate the QUEFTS model and the deviation between the measured and simulated data. The SPSS 18.0 software (SPSS Inc., Chicago, IL, USA) was utilized to analyze the significant differences between the means

of simulated and observed values using the *t*-test at the 5% significance level. RMSE and n-RMSE were calculated using the following formula:

$$\text{RMSE} = \sqrt{\frac{\sum_{i=1}^{n}(s_i - m_i)^2}{n}} \tag{1}$$

$$\text{Normalised RMSE} = \frac{\text{RMSE}}{\overline{m}}$$

where $s_i$ and $m_i$ represent simulated and measured values, respectively; $n$ represents the number of measures; and represents the mean data measurement [42].

## 3. Results

### 3.1. Characteristics of Cotton Yield and Nutrient Uptake Data

Overall, the average cotton seed yield was 4274 kg/ha with a range of 1085 to 7995 kg/ha (Table 2). The average harvest index (HI) was 0.40, with a range of 0.21 to 0.59. The average concentrations of N and P in seed were 18.9 g/kg and 3.9 g/kg, ranging from 10.1 to 35.0 g/kg and 2.0 to 8.0 g/kg, respectively, and were higher than the straw at 7.4 g/kg and 1.9 g/kg, ranging from 4.0 to 11.6 g/kg and 0.1 to 5.0 g/kg, respectively. Meanwhile, the average K content in the seed was 7.5 g/kg, ranging from 0.6 to 19.0 g/kg, and was lower than in straw at 16.8 g/kg, ranging from 8.2 to 21.9 g/kg (Table 2). The average total plant uptake of N, P, and K were 132.4 kg/ha, 28.8 kg/ha, and 142.3 kg/ha, respectively, with uptake ranges of 23.1 to 363.3 kg/ha, 7.9 to 93.3 kg/ha, and 38.3 to 562.6 kg/ha. The variability of the nutrient distributions reflected the diversity in environmental conditions and tillage treatments. The average HI of N, P, and K were 0.62, 0.59, and 0.23 kg/kg, respectively (Table 2), and while approximately 62% and 59% of plant N and P, respectively, were stored in the cotton seed, 77% of K was stored in straw.

**Table 2.** Analysis of all cotton yield and nutrient uptake data in Xinjiang of China from 1996 to 2019.

| Parameter | Unit | N [1] | Mean | SD [2] | Minimum | 25%Q [3] | Median | 75%Q | Maximum |
|---|---|---|---|---|---|---|---|---|---|
| Seed yield | kg/ha | 3805 | 4274 | 1031 | 1085 | 3611 | 4335 | 4868 | 7995 |
| Straw yield | kg/ha | 3408 | 6591 | 2120 | 1627 | 5139 | 6510 | 7784 | 21,556 |
| Harvest Index | kg/kg | 3408 | 0.40 | 0.04 | 0.21 | 0.37 | 0.40 | 0.42 | 0.59 |
| N in seed | g/kg | 3136 | 18.9 | 5.2 | 10.1 | 14.7 | 16.2 | 23.3 | 35.0 |
| P in seed | g/kg | 3136 | 3.9 | 0.7 | 2.0 | 3.3 | 3.7 | 4.3 | 8.0 |
| K in seed | g/kg | 3136 | 7.5 | 1.1 | 0.6 | 5.4 | 7.2 | 9.1 | 19.0 |
| N in straw | g/kg | 3136 | 7.4 | 1.2 | 4.0 | 6.7 | 7.3 | 8.1 | 11.6 |
| P in straw | g/kg | 3136 | 1.9 | 0.6 | 0.1 | 1.4 | 1.9 | 2.4 | 5.0 |
| K in straw | g/kg | 3136 | 16.8 | 2.0 | 8.2 | 15.5 | 16.5 | 18.0 | 21.9 |
| N uptake seed | kg/ha | 3302 | 83.2 | 34.8 | 13.9 | 58.6 | 73.1 | 104.4 | 263.9 |
| P uptake seed | kg/ha | 3252 | 16.7 | 5.6 | 4.2 | 13.1 | 15.9 | 19.8 | 55.1 |
| K uptake seed | kg/ha | 3292 | 32.8 | 15.2 | 1.7 | 23.1 | 30.0 | 40.0 | 154.0 |
| N uptake straw | kg/ha | 3302 | 49.3 | 19.8 | 9.2 | 36.5 | 47.1 | 58.3 | 248.0 |
| P uptake straw | kg/ha | 3252 | 12.1 | 5.2 | 0.2 | 8.3 | 11.1 | 15.2 | 55.4 |
| K uptake straw | kg/ha | 3292 | 109.5 | 36.1 | 28.5 | 84.2 | 107.5 | 129.7 | 416.2 |
| N uptake total | kg/ha | 3302 | 132.4 | 47.2 | 23.1 | 99.8 | 123.1 | 157.5 | 363.3 |
| P uptake total | kg/ha | 3252 | 28.8 | 8.9 | 7.9 | 22.8 | 28.0 | 33.9 | 93.3 |
| K uptake total | kg/ha | 3292 | 142.3 | 45.8 | 38.3 | 111.9 | 139.5 | 167.3 | 562.6 |
| N-HI [4] | kg/kg | 3302 | 0.62 | 0.09 | 0.12 | 0.57 | 0.62 | 0.68 | 0.86 |
| P-HI | kg/kg | 3252 | 0.59 | 0.10 | 0.31 | 0.50 | 0.58 | 0.67 | 0.98 |
| K-HI | kg/kg | 3292 | 0.23 | 0.07 | 0.02 | 0.18 | 0.22 | 0.27 | 0.68 |

[1] number of observations; [2] standard deviation; [3] 25%Q, Med., and 75%Q represent the 25th, 50th, and 75th percentiles of the database, respectively; [4] N-HI, nutrient harvest index of N; P-HI, nutrient harvest index of P; K-HI, nutrient harvest index of K.

### 3.2. Characteristics of Internal Efficiency (IE)

Internal efficiency (IE) and reciprocal internal efficiency (RIE) were used to estimate the relationship between cotton seed yield and nutrient uptake in the above-ground dry matter. In our dataset, the average IE of N, P, and K were 34.0, 152.8, and 31.1 kg/kg, ranging from 15.4 to 55.4 kg/kg for N, 65.5 to 302.2 kg/kg for P, and 31.1 to 54.0 kg/kg for K, respectively. The average RIE (total plant nutrient requirement above-ground to produce 1000 kg of seed yield) N, P, and K needed were 30.8, 6.7, and 33.3 kg ranging from 18.1 to 65.1 kg for N, 3.3 to 15.3 kg for P, and 18.5 to 82.2 kg for K, respectively (Table 3).

**Table 3.** Characteristic of internal efficiency (IE) and reciprocal internal efficiency of cotton in Xinjiang, China.

| Parameter | Unit | N [1] | Mean | SD [2] | Minimum | 25%Q [3] | Median | 75%Q | Maximum |
|---|---|---|---|---|---|---|---|---|---|
| IE-N | kg/kg | 3302 | 34.0 | 7.0 | 15.4 | 28.2 | 34.1 | 39.4 | 55.4 |
| IE-P | kg/kg | 3252 | 152.8 | 26.4 | 65.5 | 135.0 | 150.5 | 168.9 | 302.2 |
| IE-K | kg/kg | 3292 | 31.1 | 5.7 | 12.2 | 27.1 | 30.7 | 34.6 | 54.0 |
| RIE-N | kg/t | 3302 | 30.8 | 7.0 | 18.1 | 25.4 | 29.4 | 35.5 | 65.1 |
| RIE-P | kg/t | 3252 | 6.7 | 1.2 | 3.3 | 5.9 | 6.6 | 7.4 | 15.3 |
| RIE-K | kg/t | 3292 | 33.3 | 6.4 | 18.5 | 28.9 | 32.6 | 37.0 | 82.2 |

[1] Number of observations; [2] standard deviation; [3] 25%Q, Med., and 75%Q represent the 25th, 50th, and 75th percentiles of the database, respectively.

We used the QUEFTS model to define the two linear functions: the maximum accumulation (a) and maximum dilution (d) of N, P, and K in cotton. Based on previous studies [35,38,43], 2.5th and 97.5th of the IE are defined as the values of a and d. In addition, to test the feasibility of this dataset on cotton, another two datasets for the model were estimated by excluding the upper and lower 5.0th and 7.5th percentiles (SetII and SetIII, respectively) of all IEs for cotton (Table 4).

**Table 4.** Characteristic of internal efficiency (IE) and reciprocal internal efficiency of cotton in Xinjiang, China.

| Nutrients | SetI | | SetII | | SetIII | |
|---|---|---|---|---|---|---|
| | a(2.5th) | d(97.5th) | a(5th) | d(95th) | a(7.5th) | d(92.5th) |
| N | 22.3 | 46.6 | 23.4 | 44.9 | 24.3 | 43.9 |
| P | 106.4 | 208.9 | 113.9 | 200.4 | 117.8 | 194.0 |
| K | 21.2 | 44.5 | 22.7 | 42.0 | 23.7 | 40.1 |

In the table, the maximum accumulation (a), and maximum dilution (d) were calculated by excluding the upper and lower 2.5th, 5.0th, and 7.5th percentile of all measured IE.

The potential seed yield of cotton at 8000 kg/ha was calculated based on SetI, SetII, and SetIII, and presented in Figure 2. SetI was used to estimate the relationship between seed yield and nutrient requirements due to the larger range of variability. For cotton, the values of a and d were 22.3 and 46.6 for N, 106.4 and 208.9 for P, and 21.2 and 44.5 for K, respectively.

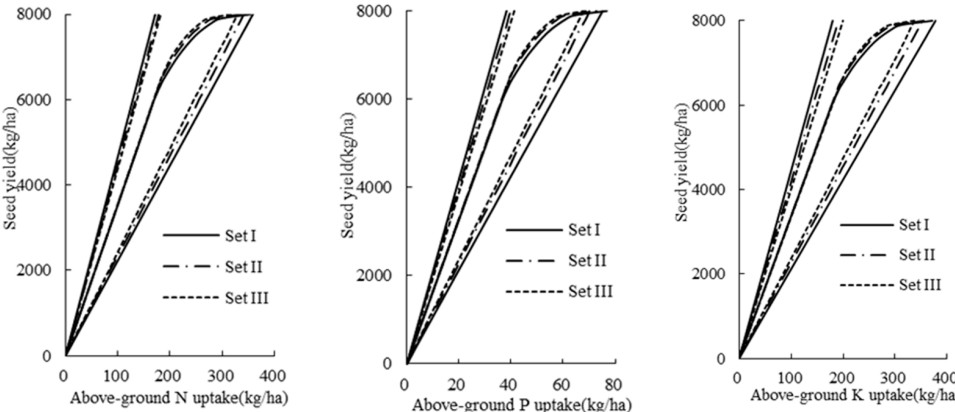

**Figure 2.** Relationships between seed yield and nutrient uptake of cotton at different sets of constants for a and d. SetsI, II, and III excluded the upper and lower 2.5th, 5.0th, and 7.5th percentiles of all internal efficiency (IE) data, respectively. YD, YA, and YU represented the maximum dilution, maximum accumulation and balanced uptake of N, P, and K in plant dry matter, respectively. The potential cotton seed yield was set at 8000 kg/ha.

*3.3. Estimation of Nutrient Uptake Requirements*

The balanced nutrient requirements were estimated by the QUEFTS model under different target yields from 4000 to 8000 kg/ha with the SetI *a* and *d* values. Since the average cotton yield in Xinjiang was between 1085 and 7995 kg/ha, a potential yield of 8000 kg/ha was used to simulate the plant N, P, and K uptakes, and the cotton yield rarely exceeded this level in Xinjiang, China. The model estimated a linear relationship between target yield and optimum nutrient uptake until the target yield reached approximately 60 to 70% of the potential yield (Figure 3). When the target yield exceeded 70% of the potential yield, more nutrients were required to increase yield (meaning a decline of IE) until the yield target approached potential yield. There were great differences in the balanced N, P, and K uptake requirements for targeted yields because of the potential yields.

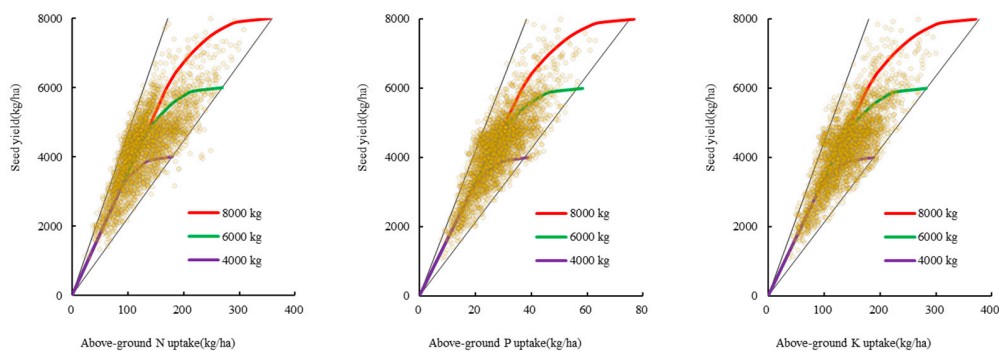

**Figure 3.** Relationships between cotton seed yields and the N, P, and K uptake in the total above-ground plant matter at maturity under different potential yields, predicted by the quantitative evaluation of fertility of tropical soils (QUEFTS) model. These parameters were calculated by the QUEFTS model after excluding the upper and lower 2.5th percentiles of all the internal efficiency data (HI range: 0.21 to 0.59; and potential cotton seed yield range: 4000 to 8000 kg/ha).

The QUEFTS model simulated balanced nutrient uptake requirements to produce 1000 kg of seed cotton were 28.3 kg N, 6.1 kg P, and 29.6 kg K when the yield reached about 60–70% of the target yield, and corresponding IE values were 35.4 kg N, 163.2 kg P, and 33.7 kg K. The optimal N:P:K ratios were 4.62:1:4.84 for total cotton plants and 4.71:1.0:1.77

for seed, which were similar to the ratios of 4.49:1.0:4.91 mean measured N:P:K uptake from filed experiments.

Seed nutrient requirements were also simulated by the QUEFTS model, as the nutrients being removed in harvested cotton seeds (or other portions of the plant) needed to be accounted for to provide guidance for the application of the appropriate amount of fertilizer to avoid waste and overfertilization [42]. The balanced nutrients of seeds by the QUEFTS model under different target yields were consistent with the nutrient uptake of the whole plant, which displayed a linear relationship until the target yield reached approximately 60 to 70% of the potential yield (Table 5). Seed nutrient uptake requirements to produce 1000 kg of seed cotton were 17.8 kg N, 3.8 kg P, and 6.7 kg K when the yield reached about 60–70% of the target yield, and the corresponding IE values were 56.2 kg N, 263.9 kg P, and 149.4 kg K.

**Table 5.** Internal efficiency (IE), the reciprocal internal efficiency (RIE) of N, P, and K in the above-ground and seed cotton, and removal ratio at different target yields simulated by the QUEFTS model for cotton in Xingjiang, China, with a potential yield of 8000 kg/ha.

| Seed Cotton Yield (kg/ha) | Above-Ground IE (kg/ha) [1] | | | Above-Ground RIE (kg/t) [2] | | | Seed Cotton RIE (kg/t) [3] | | | Nutrients Harvest Index (kg/kg) [4] | | |
|---|---|---|---|---|---|---|---|---|---|---|---|---|
| | N | P | K | N | P | K | N | P | K | N | P | K |
| 500 | 35.4 | 163.2 | 33.7 | 28.3 | 6.1 | 29.6 | 17.8 | 3.8 | 6.7 | 0.63 | 0.62 | 0.23 |
| 1000 | 35.4 | 163.2 | 33.7 | 28.3 | 6.1 | 29.6 | 17.8 | 3.8 | 6.7 | 0.63 | 0.62 | 0.23 |
| 2000 | 35.4 | 163.2 | 33.7 | 28.3 | 6.1 | 29.6 | 17.8 | 3.8 | 6.7 | 0.63 | 0.62 | 0.23 |
| 2500 | 35.4 | 163.2 | 33.7 | 28.3 | 6.1 | 29.6 | 17.8 | 3.8 | 6.7 | 0.63 | 0.62 | 0.23 |
| 3000 | 35.4 | 163.2 | 33.7 | 28.3 | 6.1 | 29.6 | 17.8 | 3.8 | 6.7 | 0.63 | 0.62 | 0.23 |
| 3500 | 35.4 | 163.2 | 33.7 | 28.3 | 6.1 | 29.6 | 17.8 | 3.8 | 6.7 | 0.63 | 0.62 | 0.23 |
| 4000 | 35.4 | 163.2 | 33.7 | 28.3 | 6.1 | 29.6 | 17.8 | 3.8 | 6.7 | 0.63 | 0.62 | 0.23 |
| 4500 | 35.4 | 163.2 | 33.7 | 28.3 | 6.1 | 29.6 | 17.8 | 3.8 | 6.7 | 0.63 | 0.62 | 0.23 |
| 5000 | 35.4 | 163.2 | 33.7 | 28.3 | 6.1 | 29.6 | 17.8 | 3.8 | 6.7 | 0.63 | 0.62 | 0.23 |
| 5500 | 35.4 | 163.2 | 33.7 | 28.3 | 6.1 | 29.6 | 17.8 | 3.8 | 6.7 | 0.63 | 0.62 | 0.23 |
| 6000 | 35.4 | 163.2 | 33.7 | 28.3 | 6.1 | 29.6 | 17.8 | 3.8 | 6.7 | 0.63 | 0.62 | 0.23 |
| 6500 | 34.4 | 158.9 | 32.8 | 29.1 | 6.3 | 30.4 | 18.4 | 3.9 | 6.9 | 0.63 | 0.62 | 0.23 |
| 7000 | 32.6 | 150.6 | 31.1 | 30.7 | 6.6 | 32.1 | 19.5 | 4.1 | 7.3 | 0.64 | 0.62 | 0.23 |
| 7500 | 30.2 | 139.3 | 28.8 | 33.1 | 7.2 | 34.7 | 21.1 | 4.5 | 7.9 | 0.64 | 0.63 | 0.23 |
| 8000 | 22.6 | 104.2 | 21.5 | 44.3 | 9.6 | 46.4 | 28.1 | 6.0 | 10.6 | 0.64 | 0.62 | 0.23 |

[1] Amount of cotton seed yield produced per unit of nutrient uptake in the plant dry matter; [2] Expressed as kilograms of total plant nutrient requirement to produce per ton of cotton seed yield; [3] Expressed as kilograms of seed nutrient requirements to produce per ton of cotton seed yield; [4] Expressed as the ratios of nutrient in the seed of the total plants.

*3.4. Field Validation*

We analyzed the relationship between observed and simulated nutrient uptake data for cotton field experiments in 2017–2020 to validate the QUEFTS model. RMSE and n-RMSE were used to evaluate the QUEFTS model. The values of RMSE were 43.3, 10.4, and 54.9 kg/ha for N, P, and K, respectively. The n-RMSE values were 20.2%, 21.2%, and 26.2% for N, P, and K, respectively. While there was some deviation for P and K, the observed and simulated N, P, and K uptake in the above-ground parts occurred near the 1:1 line, and the *p* values for N, P, and K were 0.755, 0.637, and 0.504, respectively, observed and simulated nutrient uptake values were similar, with no significant difference (Figure 4). In validation trials, organic fertilizer combined with inorganic fertilizer application NE (10% OF) significantly decreased nutrient uptake and cotton seed yield compared to inorganic fertilizer application NE in low fertility soil. This was due to the fact that organic fertilizer replacing inorganic fertilizer in low fertility soil could not meet the nutrient requirement of cotton growth. We could cut down 10% N of the currently inorganic fertilization rate with a combination of organic fertilizer applications to achieve robust production and economic returns in the future in high fertility soil, according to these balanced nutrient requirements for cotton production.

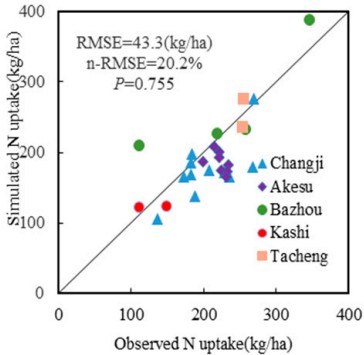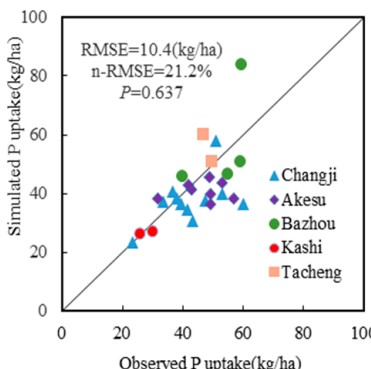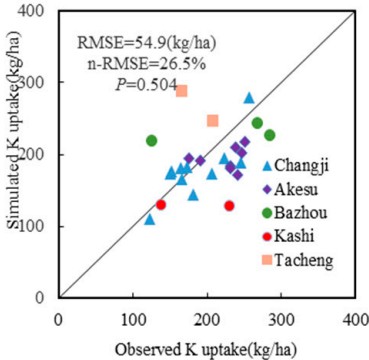

**Figure 4.** Comparisons of the simulated and observed data of the N, P, and K uptake in the total cotton dry plant matter between 2017 and 2019. The observed data were obtained from the experimental field plots where a Nutrient Expert (NE) Decision Support System was applied; the simulated nutrient uptake data were estimated using the QUEFTS model.

## 4. Discussion

In the present study, there were considerable variations in cotton seed yield and nutrient concentrations of straw and cotton seed. For cotton seed yield, the variation was likely due to the different climate types, soil fertility, cotton varieties, and nutrient management measures of the respective regions [5,44–46]. Average cotton seed yield in the current database was found to be higher than the 4881 kg/ha in China's mainland and the 176 kg/ha for the world [2], reflecting the cotton varieties, better field management techniques, and more favorable soil nutrient and climate conditions in Xinjiang [1]. Among the nutrient concentrations, the lower data were mainly from omission plots for N, P, and K, while the higher data were derived from high fertilization rates (Table 2). The differences in nutrient concentrations in seed cotton and straw yield were the result of variation in nutrient uptake in the shoot. This high variability in cotton seed yield and nutrient uptake provided a wide variation of the observed IE values in the current cotton database. Compared with the research conducted by Mao (2019) [47] (IEs of N, P, and K were 28.1, 5.38, and 22.4 kg/kg) in Shandong Province, China, the present study had higher IEs. Producing 1000 kg of cotton seed, we needed lower N and P than Zhang et al. [48]. In addition, Long suggested that 1000 kg cotton seed yield required 24.6~41.5 kg N, 5.6~13.4 kg P, and 24.2~48.9 kg K to meet the yield level of 7200~7600 kg/ha [49]. The differences in the inputs of N, P, and K might be due to the diversity of potential yields at the different experimental sites. However, overfertilization and high soil nutrient availability led to an increasing rate of higher RIEs with lower yields, confirming the presence of luxury consumption throughout the cotton growth cycle [50]. Soil nutrient conditions, plant nutrient demands, and the interactions between N, P, and K must be taken into consideration when proposing a fertilizer recommendation [40]. Therefore, a more reliable, quantitative and model simulation method is needed to estimate the optimal nutrient requirements of cotton under different target yields.

A prerequisite for using the database was a dataset where crop growth was not subjected to limitations other than in N, P, or K supply [19]. Harvest index (HI) is a complex trait signifying the balance of genetic and environmental factors leading to improved yield [51]. The harvest index (HI) for the entire cotton database ranged from 0.21 to 0.59, with an average value of 0.40, which was less than grain crops [21,28] and similar to oil rapeseed and soybean [35,52]. The 94.4% of data were centralized and distributed in the range of the 2.5th (0.32) to 97.5th (0.50) percentiles for the HI of the present cotton database.

The QUEFTS model proved useful to assess balanced nutrition, derive optimum fertilizer requirements for target yields, and explore diversity among different sites based on the large number of field trial data [53]. Determination of the boundary lines of maximum accumulation (a) and maximum dilution (d) of nutrients was one of the most critical parameters

in the generation of the cotton model. The maximum accumulation and maximum dilution of nutrients were not only influenced by variety, climate conditions, soil type, etc. but also affected by nutrient management [18,30]. In using previous studies on grain crops, radish, and sweet potato [54,55], as a guide, the borderlines of the maximum nutrient accumulation (a) and the maximum nutrient dilution (d) of cotton N, P, and K were calculated by removing 2.5th, 5.0th, and 7.5th of the IE value, respectively. The results showed that the nutrient requirements simulated by the QUEFTS model were similar for all three sets, except for differences in target yields approaching the potential yield, which was consistent with the results of previous studies [18,54]. Since the two boundaries included a larger range when the upper and lower 2.5th IE value were removed, the *a* and *d* values corresponding to the upper and lower 2.5th IE value were used as parameters of the QUEFTS model to estimate nutrient uptake of seed cotton.

The IEs and RIEs simulated by the model were the same until the targeted yield reached 60–70% of the yield potential, where plant growth was mainly limited by nutrient availability. Nutrient requirements increased, and IEs reduced drastically from the linear level when target yields approached the potential yield, which was consistent with the results of earlier studies [54–57]. On the basis of the data settings, during the constant slope increasing phase, the balanced nutrient uptake requirements to produce 1000 kg of seed cotton were 28.3 kg N, 6.1 kg P, and 29.6 kg K, and the corresponding optimal IE values were 35.4 kg N, 163.2 kg P, and 33.7 kg K for balanced nutrition. In addition, the nutrient HI calculated by the QUEFTS model (Table 5) was close to the measured values (Table 3). The results confirmed that the QUEFTS model could be used to predict the nutrient uptake of cotton.

Field validation results showed that the observed nutrient uptake was distributed closely around the 1:1 line (Figure 4), suggesting that the simulated data were consistent with the observed data, which was in line with 1:1. In addition, no significant differences were observed between the observed and simulated data ($p > 0.05$). The result confirmed that the QUEFTS model could be used to predict nutrient requirements and fertilizer requirements for cotton cultivation.

In cotton production, excessive and unbalanced nutrient levels resulted in low nutrient use efficiency. The results of this study showed that it was more beneficial to maximize nutrient utilization rate by the balanced application of N, P, and K nutrients than to pursue a higher target yield close to the potential yield. Therefore, improving nutrient internal efficiency through balanced fertilization is the key measure to achieving high yield and high efficiency of cotton production in Xinjiang, China.

## 5. Conclusions

The nutrient uptake requirements for cotton were stimulated by a QUEFTS model under 4000~8000 kg/ha potential yields. Regardless of yield potential, the model predicted a linear increase in yield if nutrients were taken up in balanced amounts of 28.3 kg N, 6.1 kg P, and 29.6 kg K per 1000 kg of seed cotton production until yield reached about 60–70% of the yield potential, with an N:P:K ratio of 4.62:1:4.84. The corresponding IEs for N, P, and K were 35.4, 163.2, and 33.7 for balanced nutrition. The optimal N, P, and K removals in 1000 kg cotton seed were 17.8, 3.8, and 6.7 kg, respectively, accounting for 62.8%, 62.3%, and 22.6% of the N, P, and K in shoots, respectively. Field validation indicated that the QUEFTS model could be used to estimate nutrient uptake at a targeted yield and contribute to developing a rational nutrient management practice strategy for Xinjiang cotton production.

**Author Contributions:** Data curation and investigation, H.Y., Q.L. and Y.Z.; Formal analysis, H.Y. and X.X.; Writing–original draft, H.Y.; Design, supervision study, revision and approvval of manuscript for publication, Y.Z. and P.H. All authors have read and agreed to the published version of the manuscript.

**Funding:** This study was funded by the National Key Research and Development Project of China (2016YFD0200102).

**Institutional Review Board Statement:** Not applicable.

**Informed Consent Statement:** Not applicable.

**Data Availability Statement:** The data presented in this study are available upon request from the corresponding author. The data are not publicly available, due to shared ownership between allparties that contributed to the research.

**Conflicts of Interest:** The authors declare no conflict of interest.

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
