# Peer review of "Estimation of Cotton Nutrient Uptake Based on the QUEFTS Model in Xinjiang"

_agronomy, doi:10.3390/agronomy12061427_

Round 1
Reviewer 1 Report
Dear Authors,
It is an interesting attempt to propose a model for determining the balanced nutritional needs of cotton.
However, the use of data from older works and not from experiments designed specifically for the purpose of creating a model, results in excessive variability in the results, which is not treated in a statistically correct approach and contradict similar results in the literature, for a subject very well studied like the nutrition of cotton.
In addition to the comments and remarks that I have incorporated in the text I would like to fill in as general comments that
- the introduction needs improvement to pose a specific problem that is the purpose of the work.
- The methods and materials need careful corrections as parameters that are not used later are reported, and vice versa. Authors should be careful with terminology because they are often confusing.
- Results should not be presented with commentary as there is a separate section.
- Discussion of the results is very small and without sufficient and recent bibliographical references,
- conclusions do not answer the introductory questions

Author Response
Comments to the Author(1)It is an interesting attempt to propose a model for determining the balanced nutritional needs of cotton. However, the use of data from older works and not from experiments designed specifically for the purpose of creating a model, results in excessive variability in the results, which is not treated in a statistically correct approach and contradict similar results in the literature, for a subject very well studied like the nutrition of cotton.In addition to the comments and remarks that I have incorporated in the text I would like to fill in as general comments that.
Answer: Thanks for your comments. We have added sentences to discuss the inconsistent results of our findings and previous studies, please see of the revised version.
(1)The introduction needs improvement to pose a specific problem that is the purpose of the work.Answer: We have rewrite our introduction to make it more clear and comprehensible. Also, the English grammar are checked by an American friend.
(2)The methods and materials need careful corrections as parameters that are not used later are reported, and vice versa. Authors should be careful with terminology because they are often confusing.Answer: Sorry for the confusion. We have correct methods and materials parts, Please see the revised manuscript.
(3)Results should not be presented with commentary as there is a separate section.Answer: We have separate the results parts from the discussion. Please see the revised manuscript.
(4)Discussion of the results is very small and without sufficient and recent bibliographical references.Answer:Thanks for your comments, we have rewrite our discussion .Please see the revised manuscript.
(5)Conclusions do not answer the introductory questionsAnswer:Thanks for your comments, we have perfect our conclusions .Please see the revised manuscript.
(6)Page 1, Line 30: check grammer ” producers”
Answer: “producers” has been replaced by “producer” in line 29 of the revised manuscript.
(7)Page 1-2, Line 43-46 : please rephrase
Answer: We have rephrase this sentence to make it clear:” These methods, while effective for fertilizer recommendation, generally only involve a single nutrient, and the interactions among N, P and K are not considered. As soil and plant chemical analyses are both time-consuming and expensive, regional scale fertilization has generally proceeded without them.” Please see line 49-52 of the revised manuscript.
(8)Line 48: check grammer “provide”
Answer: “provide” has been replaced by “providing” in line 54 of the revised manuscript.
(9)Line 50-51: please rephrase
Answer: We have rephrase this sentence to make it clear:” The QUEFTS model was modified by Smaling and Janssen (1993), to estimate the nutrient requirements for a specific yield target, and determine the required types and amounts of fertilizer.” Please see line 56-58 of the revised manuscript.
(10) Line 62:???
Answer: Sorry for the confusion. It was “Develop functions”. Please see line 74 of the revised manuscript.
(11)Line 78-79 :please rephrase
Answer: We have rephrase this sentence to make it clear:” covering 18 cotton-growing counties and 61 cotton varieties in the Xinjiang region of China.” Please see line 90-91 of the revised manuscript.
(12)Line 87 : please correct
Answer: Sorry, We have correct this sentence. Please see line 107 of the revised manuscript.
(13)Line 107,108 :please rephrase
Answer: We have rephrase this sentence to make it clear:” The plot sizes of each experiment were 43.2 m2 and distributed in a randomized block design.” Please see line 128-129 of the revised manuscript.
(14)Line 118:check grammer “were ranged ”
Answer: “were ranged” has been replaced by “ranged” in line 140 of the revised manuscript.
(15)Line 123,124 :check grammer
Answer: We have revised this sentence as “K fertilizer potassium sulfate (50% K2O) was applied 50% at ray stage, and 50% at flowering stage.” Please see the revised version in line 145-146.
(16)Line 137 : please rephrase
Answer: We have deleted this word from the revised manuscript. Please see the revised version in line 159.
(17)Line 152-154,156-157:this is discussion
Answer:We have removed this sentemse from results to discussion parts in the revised manuscript. Please see the revised manuscript.
(18)Line 155,173 :HI and RIE is not described in M&M section
Answer: We have described HI , IE and RIE in M&M section. Please see the the revised version in line 93-98.
(19)Table 2:check if numbers are correct. eg 4274*18.9=80.8
Answer: Thanks very much for all your careful and patient comments . Not every experimental point has nutrient content. For example, the data from the published field experiments in the peer reviewed journal were only nutrient uptake, so the nutrient data are not left by yield, dry matter and nutrient content.
(20) Line 75-177:in lines 92-93 you describe IE for seed not for lint. Why do you compare such different production parts of cotton?
Answer: Sorry. We have correct this parts, and add corresponding bibliography. Please see the the revised version in line 305-315.
(21) These studies refer to an IE calculated in a different way. Please justify the use in your research.
Answer: Actually, we used the same way to calculate IE as these previous studies. Sorry for the confused sentence, we have revised line 97 as:“cotton seed yield per unit of nutrient uptake in the above-ground parts”.
(22) Line 187-is this seed + lint? If yes IE should be corrected and all calculations resulting from IE
Answer: Sorry for the confusion. It was “seed yield”. Please see line 215 of the revised manuscript.
(23) Line 216-217-NPK ratio needs for cotton is very well studied. Typically is 2:1:1. Your findings could be a result of the extremely high variation you have in all measured and calculated data. A careful statistical analysis must be done an presented in the paper to ensure that use of mean values is correct and not misleading.
Answer: Thanks for your comments, we have careful statistical analysis in the revised manuscript.
(24)Line 222~224:Please elaborate or rephrase
Answer: We have rephrase this sentence to make it clear:” Seed nutrient requirements were also simulated by the QUEFTS model, as the nutrients being removed in harvested cotton seeds (or other portions of the plant) need to be accounted for to provide guidance for the application of the appropriate amount of fertilizer, to avoid waste and overfertilization.” Please see line 251-254 of the revised manuscript.
(25)Line 226:the relationship is clearly non linear for all three nutrients. Please advise literature.
Answer: Sorry, We have correct this sentence. Please see line 256 of the revised manuscript.
(26)Line 257:please correct
Answer: We applozise for this, we have correct this word. Please see line 306 of the revised manuscript.
(27)Line 257-261:Unfortunately I agree.The extremely high variation does not support the reliability of the method unless a proper statistical approach is followed and clearly stated in the manuscript.
Answer:Sorry, we didn't make it clear what we meant. We have rewriteen this parts, please seeline 306-307 of the revised manuscript.
(28)Line 264-265:Your findings are not in accordance with literature regarding cotton balanced nutrition. Hoe to you discuss it?
Answer: Sorry, we have deleted this sentence.
(29)Line 272-274:These studies use IE just for lint kg/ nutrient kg. Please be careful.
Answer: Sorry for the confusion. It was “seed yield”. Please see line 329 of the revised manuscript.
(30)Line 282-According line 151 seed cotton yield fluctuated from 1000 to 8000 kg/ha.
Later on you used potential yields from 4000 to 8000. Which is the potential yield?
Answer: Potential yield is defined as that which is determined by the amount of incident solar radiation, CO2 concentration, temperature, and plant density for a given crop variety or hybrid in a specific growth environment. Limitations to the attainment of this optimum potential yield could result from changes in natural determinants of crop productivity such as rainfall variability, nutrient decline, pest infestation, and weed invasion. Since the average cotton yield in Xinjiang was among 1085 to 7995 kg/ ha, a potential yield of 8000 kg/ha was used to simulate the plant N, P, and K uptakes and the cotton yield rarely exceeded this level in Xinjiang.

Author Response
Comments to the Author
Your paper deals with a topic of great interest that falls within the aim of the Journal. Anyway, the manuscript has many flaws and needs MAJOR revision.
My greatest concern is that the discussion is not strongly supported and wrongly presented.
Specific suggestions to improve the submission are given below.
(1)Line 93:The steps of the QUEFTS model have not been described in this paper. The authors just cited the corresponding bibliography. A brief report of the model should be provided in Introduction or in Materials and Methods
Answer: We applozise for this, we have perfects steps of QUEFTS model. Please see line 112-121 of the revised manuscript.
(2)Line 134-139:The proper bibliography for the analyses used is missing
Answer: We have added bibliography for the analyses, please see line 159 of the the revised manuscript.
(3)Line 154:“Average cotton seed yield in the current database was found to be higher than the 4881 kg/ha in China’s mainland and the 176 kg/ha for the world” The meaning of this sentence is not clear; the differences are enormous. How the authors explain the differences;
Answer: Sorry for the unclear. We have explained the differences, please see line 297-301 of the the revised manuscript.
(4)Line 140:After line 140 is not Field validation and the authors should use different heading such as Statistical analysis or Measurement and calculation.
Answer: We have added heading Statistical analysis, please see line 162 of the the revised manuscript.
(5)Tables:All tables should have footer explaining the entries in the table. The headings also should be improved
Answer: Thanks for your comment. We have improved table headings and added footer explaining the entries, please see the revised version.
(6)Line 241-242:“and the P values for N, P, and K were 0.755, 0.637, and 0.504, respectively, observed and simulated nutrient uptake values were similar, with no significant difference”The authors mean p values and although they mentioned that there are no significant differences in the figure that followed p values are 0.755**, 0.637**, and 0.504** (** indicates significant differences)
Answer: We applozise for this, we have correct Figure 5. Please see the revised manuscript.
(7)Figure 4:Although in Table 1 the authors separated Xinjiang in Southern and Northern Xinjiang and in line 103 mentioned 5 regions, Figure 5 displayed/showed one unique region. It seems like 2.3 and 3.4 is not in agreement.
Answer: Sorry, We have added regions in Figure 5. Please see the revised manuscript.
(8)Discussion:Discussion is not addressing the stated objectives of this study. The discussion should support better the results presented above. There is little information showing/explaining the objectives of the study. Discussion should be improved
Answer: We have rewritten the discussion part to make it clear according to your suggestion, please see the revised version.

Round 2
Reviewer 1 Report
Dear Authors,
the revised text has clearly improved the level of English used.
Nevertheless, the important issues raised during the first revision remain. Only the suggested corrections have been made.
As for the critical questions that were asked regarding the methodology followed, the discussion of the results and the use of appropriate bibliographic references to support it, sufficient and clear answers were not given, nor were any substantial corrections, changes or additions made.
Avoiding repeating my original position in detail, please consult it more carefully.
Author Response
the revised text has clearly improved the level of English used.
Nevertheless, the important issues raised during the first revision remain. Only the suggested corrections have been made.
As for the critical questions that were asked regarding the methodology followed, the discussion of the results and the use of appropriate bibliographic references to support it, sufficient and clear answers were not given, nor were any substantial corrections, changes or additions made.
Avoiding repeating my original position in detail, please consult it more carefully.
Answer: Thanks very much for all your careful and patient comments. We have rewrite our discussion to make it clear. The data from 1996 to 2019 reflected the development process of cotton planting in Xinjiang. Earlier data are mainly yield response data, indicating Xinjiang soil fertility. For data variation was likely due to the different climate types, soil fertility, cotton varieties and nutrient management measures of the respective regions. The quantitative evaluation of the fertility of tropical soils (QUEFTS) model has been successfully applied to various crops providing recommendations for fertilizer application for a number of countries and covering a variety of soil types and agricultural environments, and our statistical method is the same as rice, wheat, soybean and other crops in China.

Reviewer 2 Report
Although most of the corrections I suggested have been made, discussion remains my great concern. You just added a simple repetition of the results without changing the discussion which needs improvement.
Author Response
Although most of the corrections I suggested have been made, discussion remains my great concern. You just added a simple repetition of the results without changing the discussion which needs improvement.
Answer: We have improved the discussion part according to your suggestion, please see the revised version.
